# Uncertainty in Aerosol Radiative Forcing Impacts the Simulated Global Monsoon in the 20th Century

Jonathan K. P. Shonk[1, *], Andrew G. Turner[1, 2], Amulya Chevuturi[1], Laura J. Wilcox[1], Andrea J. Dittus[1] and Ed Hawkins[1]

[1] National Centre for Atmospheric Science, University of Reading, Reading, UK
[2] Department of Meteorology, University of Reading, Reading, UK
[*] Current affiliation: Met Office @ Reading, University of Reading, Reading, UK

Corresponding author: Jonathan K. P. Shonk (current e-mail address: jon.shonk@metoffice.gov.uk)

**Abstract.** Anthropogenic aerosols are dominant drivers of historical monsoon rainfall change. However, large uncertainties in the radiative forcing associated with anthropogenic aerosol emissions, and the dynamical response to this forcing, lead to uncertainty in the simulated monsoon response. We use historical simulations from the "SMURPHS" project, run using HadGEM3-GC3.1, in which the time-varying aerosol emissions are scaled by factors from 0.2 to 1.5 to explore the monsoon sensitivity to historical aerosol forcing uncertainty (present-day versus preindustrial aerosol forcing in the range $-0.38$ W m$^{-2}$ to $-1.50$ W m$^{-2}$). The hemispheric asymmetry in emissions generates a strong relationship between scaling factor and both hemispheric temperature contrast and meridional location of tropical rainfall. Averaged over the period 1950–2014, increasing the scaling factor from 0.2 to 1.5 reduces the hemispheric temperature contrast by 0.9 °C, reduces the tropical summertime land–sea temperature contrast by 0.3 °C and shifts tropical rainfall southwards by 0.28° of latitude. The result is a reduction in global monsoon area by 3% and a reduction in global monsoon intensity by 2%. Despite the complexity of the monsoon system, the monsoon properties presented above vary monotonically and roughly linearly across scalings. A switch in the dominant influence on the 1950–1980 monsoon rainfall trend between greenhouse gases and aerosol is identified as the scalings increase. Regionally, aerosol scaling has a pronounced effect on Northern Hemisphere monsoon rainfall, with the strongest influence on monsoon area and intensity located in the Asian sector, where local emissions are greatest.

## 1 Background

Monsoon systems provide rainfall for billions of people, many of whom are dependent on the monsoon rains for survival. It is therefore important to understand the effects of climate change on the global monsoon, both in the past and future. Projections show a future increase in global monsoon area, rainfall amount, and rainfall intensity (Hsu et al., 2012, 2013). In contrast, studies have reported a decline in global monsoon rainfall in the latter half of the 20th century (Hsu et al., 2011; Wang and Ding, 2006; Zhou et al., 2008), particularly in Northern Hemisphere (NH) monsoons (Zhou et al., 2008).

Historical emissions of anthropogenic aerosols (AA) and their precursors cause a net negative radiative forcing, global cooling and suppression of rainfall, hence opposing the impacts of greenhouse gas (GHG) emissions (Wu et al., 2013). Furthermore,

most AA emissions arise in the NH, giving them a strong control on hemispheric temperature gradients (e.g., Wilcox et al., 2013), with profound effects on monsoon circulations (Broccoli et al., 2006; Friedman et al., 2013; Voigt et al., 2017) and interhemispheric energy and moisture transport (Haywood et al., 2016; Stephens et al., 2016). Since the middle of the 20th century, large-scale AA-driven circulation changes have acted to weaken the monsoon, and have dominated over the response to GHGs. Bollasina et al. (2011), Polson et al. (2014), Salzmann et al. (2014), Song et al. (2014) and Guo et al. (2015) have all shown that increasing AA emissions played an important part in driving regional and global monsoon rainfall decrease during the mid-20th century. However, in the future, a GHG-driven thermodynamic response is expected to dominate, driving increased monsoon rainfall (Li et al., 2015; Wilcox et al., 2020).

Land-sea contrasts affect temperature gradients and thus also affect monsoon circulation strength. On regional scales, AA-induced cooling can oppose GHG-induced warming effects (Ramanathan et al., 2005; Ramanathan and Feng, 2009), leading to a slackening of temperature contrasts between land and sea (Lau and Kim, 2017) and an increase in surface pressure (Song et al., 2014), both of which weaken the circulation. Remote AA emissions are also important, acting to change monsoon rainfall through circulation changes, albeit via different mechanisms to local AA emissions (Cowan and Cai, 2011; Dong et al., 2016; Undorf et al., 2018; Wang et al., 2017; Westervelt et al., 2018). Within regional monsoon systems, AA emissions can also change the characteristics and distribution of rainfall, affecting monsoon onset (Lau et al., 2006) and withdrawal (Guo et al., 2016).

There is substantial uncertainty in present-day top-of-atmosphere aerosol effective radiative forcing. The Fifth Assessment Report of the Intergovernmental Panel on Climate Change (IPCC) reported a 5%-to-95% confidence interval of $-1.9$ W m$^{-2}$ to $-0.1$ W m$^{-2}$ (Myhre et al., 2013), while the most recent estimate spans the range $-2.0$ W m$^{-2}$ to $-0.4$ W m$^{-2}$ (Bellouin et al., 2020). In future climate projections, in which AA emissions look likely to decrease while GHG emissions continue to increase across a range of future climate scenarios (Lund et al., 2019), the ability to capture the balance between their respective radiative effects is crucial. Future reductions in AA emissions have the potential to cause increases in global rainfall comparable to those resulting from moderate GHG increases (Rotstayn et al., 2013), with the largest increases anticipated over East and South Asia (Levy et al., 2013; Westervelt et al., 2015). Uncertainty in the magnitude of aerosol forcing, and the monsoon response to it, is compounded in climate projections where potential aerosol emission pathways are diverse. In the Asian region in particular, there is great variety in future emission trends across the Shared Socio-Economic Pathways (SSPs) and future aerosol forcing is likely to determine the magnitude of near-future changes in monsoon rainfall (Wilcox et al., 2020). Of particular importance for the monsoon is the uncertainty in the sign of the projected emission trends over China and India depending on the SSP (see Figure 1b of Samset et al., 2019).

In this study, we quantify the impact of the uncertainty in present-day aerosol radiative forcing on the global monsoon system using a set of historical climate simulations produced as part of the SMURPHS ("Securing Multidisciplinary Understanding and Prediction of Hiatus and Surge Events") project (Dittus et al., 2020). The SMURPHS ensemble consists of a set of historical climate simulations with AA emissions scaled by various factors, chosen to span the range of uncertainty in present-day aerosol effective radiative forcing. This allows us to investigate the sensitivity of historical changes in the monsoon to the

strength of aerosol forcing, without the complications arising from structural and parametric uncertainty found in a multi-model framework. The range of model biases and aerosol process representations in a multi-model ensemble, such as the most recent phase of the Coupled Model Intercomparison Project (CMIP6), preclude the attribution of differences in the response to differences in the forcing alone. We introduce the ensemble and experimental design in more detail in Section 2. The effect

5    of the aerosol scaling in terms of temperature contrasts across hemispheres, and between land and sea, is examined in Section 3. Section 4 presents the effects of scaling on standard metrics of the global and regional monsoons. We summarise and conclude in Section 5.

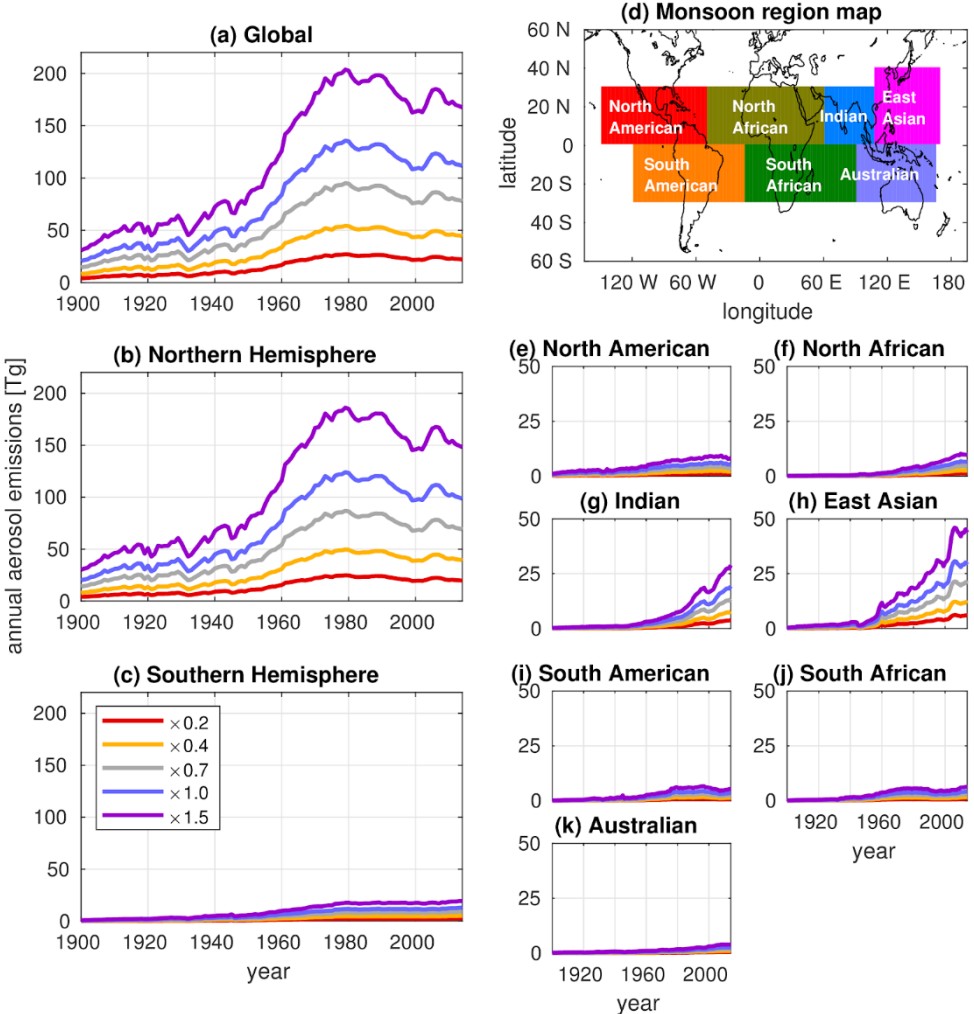

10    **Figure 1. Sulphur dioxide emissions used in SMURPHS on (a) global, (b, c) hemispheric and (e–k) regional scales, in Tg yr⁻¹. Organic and black carbon emissions are scaled in the same way. Monsoon regions are as defined in panel (d).**

## 2  SMURPHS ensemble and aerosol emission data

The SMURPHS dataset consists of historical climate simulations run over the period 1850–2014 using a fully coupled version of HadGEM3-GC3.1 at resolutions of N96 and 1° in the atmosphere and ocean respectively (Kuhlbrodt et al., 2018; Williams et al., 2018). The model version used here is a development version towards the UK submission to CMIP6 (Andrews et al, 2020), and differs only in its treatment of prescribed ozone concentrations (see Supplementary Information in Dittus et al., 2020; Hardiman et al., 2019). For the treatment of anthropogenic aerosol, HadGEM3 uses the GLOMAP two-moment aerosol scheme that includes representations of the cloud albedo and cloud lifetime effects (Mulcahy et al., 2018 and references therein). Mineral dust is simulated interactively using a bin emission scheme (Woodward, 2001). Five ensemble members are run for each of five experiments in which the historical aerosol emissions are scaled by a constant factor. This factor is applied to emissions of all species of anthropogenic aerosol and precursors, at all locations throughout the historical emissions dataset. Biomass burning emissions are included but not scaled. Five scaling factors were selected: ×0.2, ×0.4, ×0.7, ×1.0 and ×1.5, with the ×1.0 scaling corresponding to the standard CMIP6 historical protocol. The scaling factors were chosen to sample a broad range of the uncertainty in present-day aerosol radiative forcing according to Myhre et al. (2013) and Bellouin et al. (2020), and correspond to forcings of $-0.38$ W m$^{-2}$, $-0.60$ W m$^{-2}$, $-0.93$ W m$^{-2}$, $-1.17$ W m$^{-2}$ and $-1.50$ W m$^{-2}$ respectively. More detail on the SMURPHS ensemble is presented by Dittus et al. (2020).

The SMURPHS simulations use the same aerosol emission dataset as used in CMIP6 (Hoesly et al., 2018), which contains emissions from 1750–2014 for sulphur dioxide, black carbon and organic carbon. As an illustration of the time evolution of historical aerosol emissions, sulphur dioxide emissions from 1900 onwards are shown in Figure 1. In the early 20th century, emissions increased gradually, but then ramped up from 1950 to 1980. Since 1980, emission mitigation efforts in North America and Europe have been balanced by continued increases in Asia, causing global emissions to level off. The hemispheric asymmetry in AA emissions is clear, with the NH contributing approximately 90% of the global total throughout the 20th century (Figures 1b, 1c). Most monsoon regions show a gradual increase in emissions in the 20th century, with pronounced increases since 1970 seen in the Indian and East Asian sectors (Figures 1g, 1h). The Hoesly et al. (2018) emissions dataset is the most up-to-date inventory of historical AA emissions and is therefore considered the best estimate.

In this study, we use all five members from each of the five experiments, but include years from 1900 onwards, to allow 50 years for the model to adjust to the scalings (after Dittus et al., 2020). When considering climatological quantities, we consider the ensemble mean for each experiment to be the model estimate of the climate system under those scaling conditions and indicate uncertainty across ensemble members in terms of the range across the five members. Where quantities are averaged over areas, a cosine-based latitude weighting is applied.

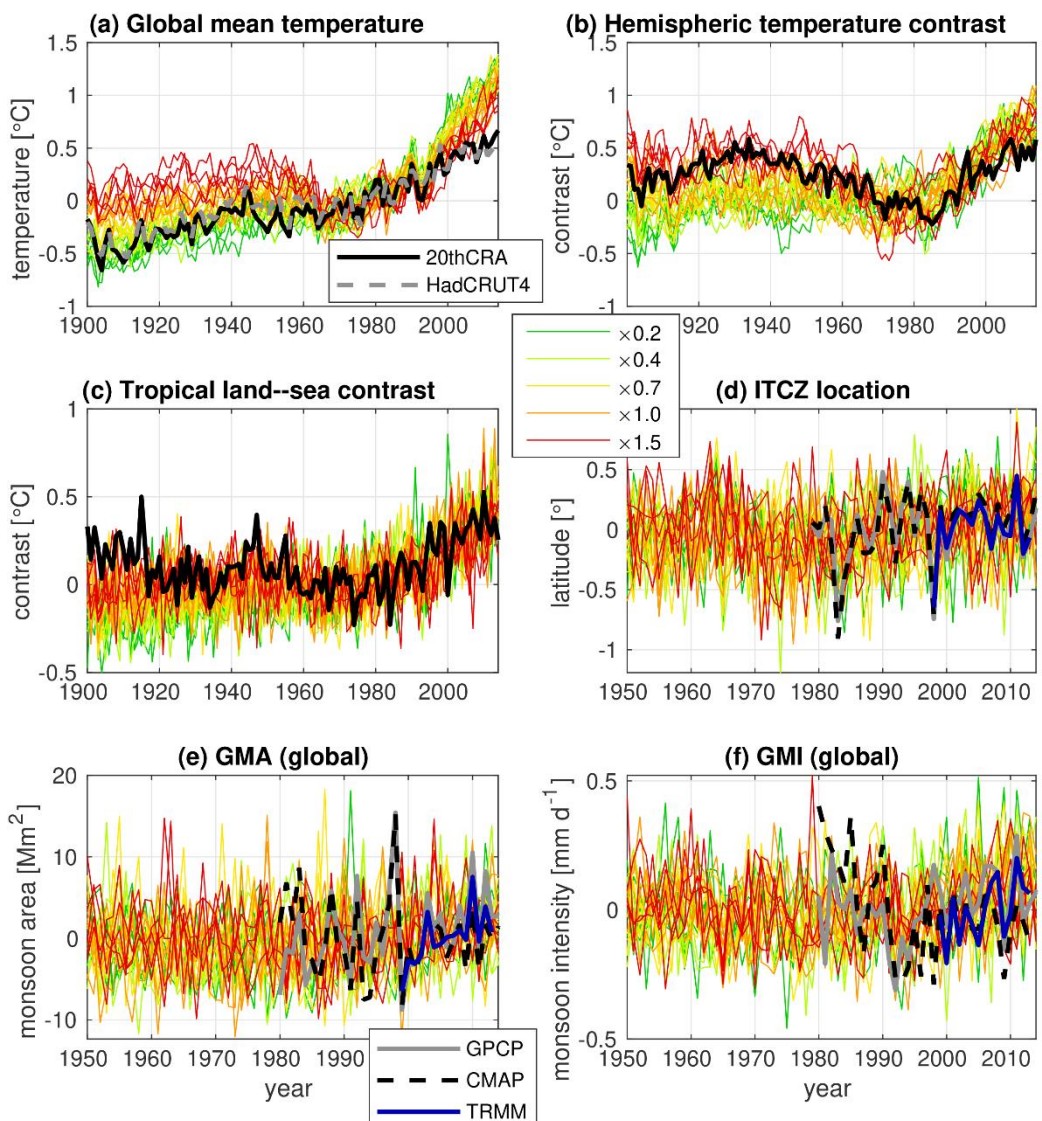

**Figure 2. Validation time series comparing the model used in the SMURPHS ensembles (HadGEM3) with observation and reanalysis datasets. For temperature quantities, we use data from 20th Century Reanalysis (20thCRA; Slivinski et al., 2019) and Hadley Centre/Climatic Research Unit Temperature (HadCRUT4; Morice et al., 2012). For rainfall quantities, we use data from the Global Precipitation Climatology Project (GPCP; Adler et al., 2003), the CPC Merged Analysis of Precipitation (CMAP; Xie and Arkin, 1997) and the Tropical Rainfall Measuring Mission (TRMM; Kummerow et al., 2000). All model ensemble members are shown, with no running means. Temperature properties span the period 1900 to 2014; rainfall properties span 1950 to 2014. Sections 3 and 4 of the paper contain descriptions of how the variables are calculated. Anomalies are calculated for each dataset and each experiment, with respect to 1961–1990 (temperature fields) and 1980–2009 (rainfall fields).**

10   The performance of HadGEM3 at representing monsoon properties used in this study is evaluated in Figure 2. The overall trends and variability in these properties compare well between model and observations with the exception of global temperature since 2000, in which the model produces an accelerated rate of warming, although this is a recognised behaviour (Dittus et al., 2020). The model also responds reliably to changing aerosol forcing, demonstrated by the dependence of

gradients in global mean temperature and hemispheric temperature contrast (HTC) in the period 1950–1980 to the magnitude of the scaling. The increasing AA emissions during this period lower global temperature, and the hemispheric asymmetry in emissions reduces the HTC. Stronger forcing scalings result in steeper declines in both properties during this period (Figures 2a, 2b). Furthermore, there is no optimal scaling factor that can reliably represent the gradients in both properties during this

period – lower scalings result in a more realistic decline of global mean temperature, while higher scalings (nearer ×1.0) generate a more realistic decline of HTC. Given the importance of HTC in influencing monsoon change (e.g., Bollasina et al., 2011), it is likely that the higher scaling factors will provide the most realistic representation of the global monsoon.

## 3   Temperatures and contrasts

The effect on global mean temperature of scaling the AA emissions is clear (Figure 3a). Higher aerosol scalings lead to cooler global temperatures, and by the 1970–2014 period there is little overlap in global temperature between scalings. We also see evidence of the control by AA emissions on the magnitude of the mid-20th-century hiatus (the period 1950–1980), in agreement with the findings of Wilcox et al. (2013) and Jones et al. (2013). The higher scalings lead to a stronger hiatus, with a global cooling of almost 0.5 °C between 1950 and 1970 in the ×1.5 experiment. The lower scalings lead to a much weaker

hiatus. In the ×0.2 experiment, there is only a brief departure from the positive temperature trend around 1960 and a hiatus is barely discernible. These results echo those of Dittus et al. (2020).

The hemispheric asymmetry of AA emissions leads to a much greater degree of cooling in the NH, so the strength of the forcing has a control on the hemispheric temperature contrast (HTC), defined as NH minus SH (Chang et al., 2011; Wilcox et al., 2013). Lower scalings reduce the degree of NH cooling and therefore result in a greater HTC, remaining at about 1 °C

from 1930 to 1990 (Figure 3b). This reflects the tendency of the NH to be, on average, warmer than the Southern Hemisphere (SH; for example, Kang et al., 2015). Under the highest scaling (×1.5), however, the HTC reduces by over 0.5 °C from 1940 to 1970, reversing in sign during the 1970s and 1980s.

This shift in HTC is reflected in the location of the ITCZ (Figure 3d). ITCZ location is determined as the latitude of the zonal mean rainfall centroid within the latitude band 20° N to 20° S, following the "centroid" method of Adam et al. (2016; "$\varphi_{cent}$"

in their Equation 1). Lower scalings, associated with a warmer NH and stronger HTC, lead to an ITCZ location that is further north, consistent with Hwang et al. (2013), Allen et al. (2015) and Chung and Soden (2017). The model places its ITCZ on the equator, in contrast to Adam et al.'s (2016) calculation using observational data, which places the ITCZ north of the equator. This is likely due to the tendency of the model to place its ITCZ rainfall too far south (Williams et al., 2018). Repeating the ITCZ location calculation using the method of Shonk et al. (2018) applied to zonal mean rainfall data shows a similar result.

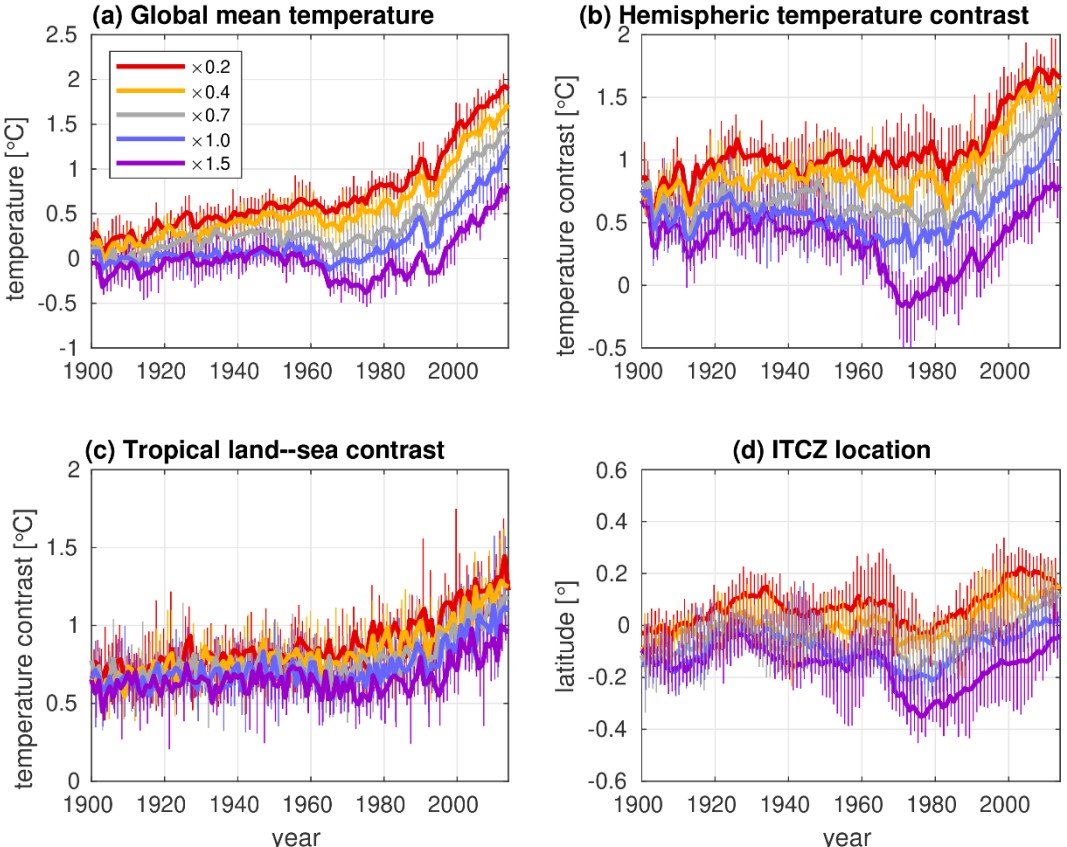

**Figure 3. Time series of various atmospheric properties from the SMURPHS simulations: (a) global mean surface air temperature, expressed as an anomaly with respect to the mean value across the five members of the ×1.0 experiment in the period 1900—1929; (b) hemispheric temperature contrast (NH minus SH); (c) tropical land–sea temperature contrast, calculated in the summer months (November–March in SH, May–September in NH) for latitudes within 30° S and 30° N only; (d) global mean ITCZ location, calculated following Adam et al. (2016; see text for a description of the method). All values are ensemble means; vertical error bars indicate the range across the five ensemble members.**

Monsoon strength is also influenced by changes in the land–sea temperature contrast (LSTC), both on regional (Lau and Kim, 2017) and global (Fasullo, 2012) scales. While weaker than the effect on HTC, there is a degree of control of the aerosol scaling on the LSTC, albeit with a larger overlap between ensemble members (Figure 3c). Higher scalings result in cooler land surfaces with respect to the surrounding oceans, hence the LSTC is reduced, and the monsoon is weakened.

The control of the aerosol forcing on the properties presented in this section is demonstrated quantitatively in the top section of Table 1 in terms of means over the 1950–2014 period, when most changes in anthropogenic aerosol have occurred. All properties vary monotonically and roughly linearly across the range of scalings used in SMURPHS, with higher scalings resulting in a cooler global temperature, a weaker HTC, an ITCZ situated further south, and a weaker LSTC. The impact of the uncertainty in present-day forcing on these properties is presented in the rightmost column of Table 1 as the differences between the lowest and highest scalings (×1.5 minus ×0.2). Changing the forcing from lowest to highest value lowers global

temperature by nearly 1 °C and reduces the HTC from 1.19 °C to 0.27 °C. The zonal-mean ITCZ location shifts southwards by 0.28° of latitude, and the LSTC reduces by just over 30%, from 0.98 °C to 0.68 °C.

**Table 1. Mean monsoon-related properties, as defined in Sections 3 and 4, averaged over the period 1950–2014 (during which global aerosol emissions increased), and all five ensemble members. The difference column is the change from ×0.2 (lowest scaling) to ×1.5 (highest scaling), expressed as a percentage where indicated.**

| | ×0.2 | ×0.4 | ×0.7 | ×1.0 | ×1.5 | Difference |
|---|---|---|---|---|---|---|
| Global mean temperature anomaly [°C] | 0.98 | 0.79 | 0.52 | 0.30 | 0.02 | **−0.95** |
| Hemispheric temperature contrast [°C] | 1.19 | 1.03 | 0.78 | 0.57 | 0.27 | **−0.91** |
| ITCZ location (latitude) [°] | 0.09 | 0.03 | −0.06 | −0.10 | −0.19 | **−0.28** |
| Tropical land–sea contrast [°C] | 0.98 | 0.92 | 0.83 | 0.77 | 0.68 | **−0.30** |
| GMA [Mm$^2$] | 126.8 | 126.0 | 125.0 | 124.3 | 122.8 | **−2.99%** |
| HMA (NH) [Mm$^2$] | 66.8 | 66.0 | 65.5 | 64.9 | 63.9 | **−4.25%** |
| HMA (SH) [Mm$^2$] | 60.0 | 59.9 | 59.5 | 59.5 | 59.0 | **−1.60%** |
| GMI [mm d$^{-1}$] | 7.76 | 7.74 | 7.69 | 7.66 | 7.61 | **−1.93%** |
| HMI (NH) [mm d$^{-1}$] | 7.68 | 7.65 | 7.57 | 7.54 | 7.48 | **−2.58%** |
| HMI (SH) [mm d$^{-1}$] | 7.86 | 7.84 | 7.82 | 7.78 | 7.76 | **−1.30%** |

## 4 Monsoon area and rainfall

We evaluate the effects of aerosol scaling on the monsoon via Global Monsoon Area (GMA) and Global Monsoon Intensity (GMI). These properties are defined following Liu et al. (2009), Hsu et al. (2011) and others: a gridbox is within the GMA if the difference in summer and winter rainfall (May to September and November to March, depending on hemisphere) is greater than 2 mm d$^{-1}$, and more than 55% of the rain falls in the summer months. The total GMA is calculated as the sum of the area of all gridboxes within the GMA region. GMI is then calculated as the total rainfall within the GMA, divided by the area of the GMA. We also define hemispheric (HMA and HMI) and regional (RMA and RMI) equivalents – these are the GMA and GMI calculated separately for either each hemisphere, or each monsoon region as defined in Figure 1d. Note that, by these definitions, monsoon regions span both land and sea. There is debate about whether monsoon indices should be defined solely over land, although here we preserve the full land-plus-sea definition of Liu et al. (2009).

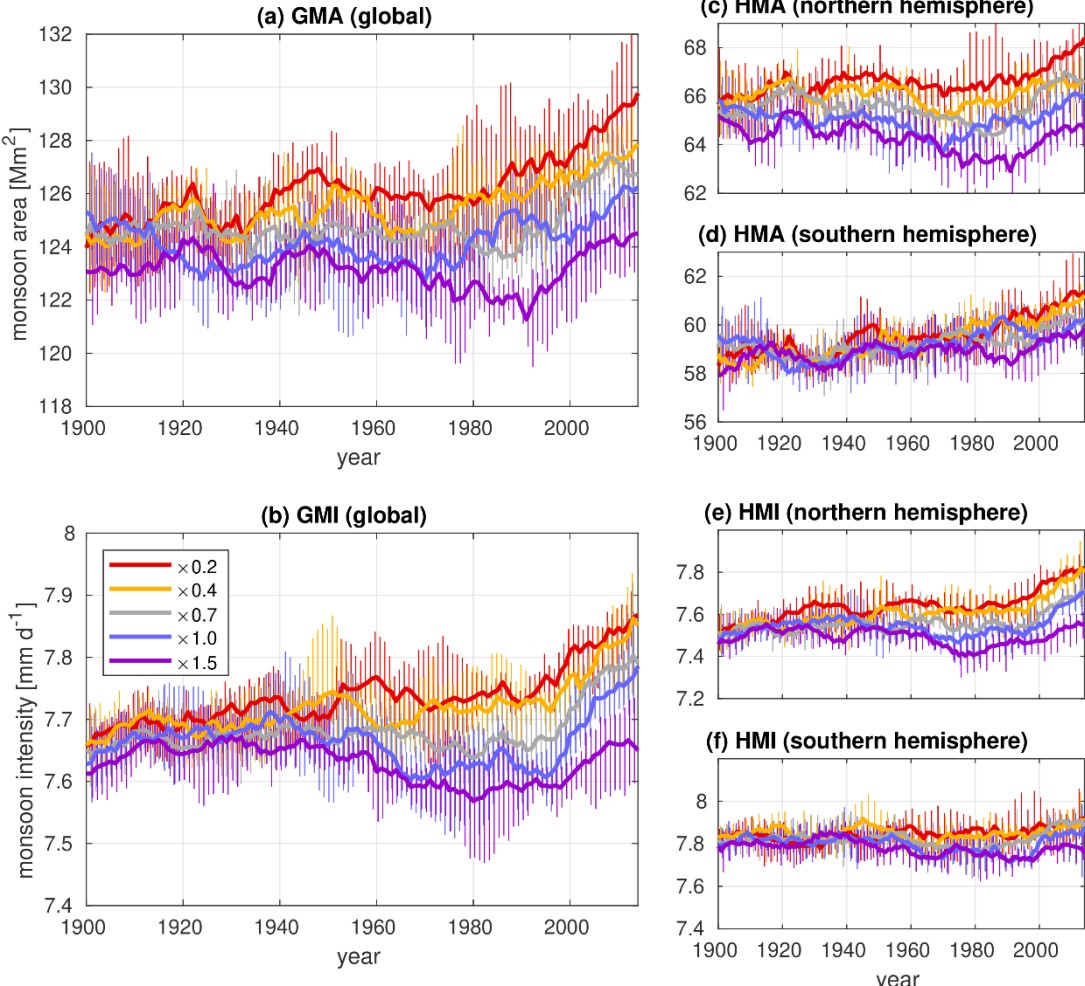

**Figure 4. Time series of (a) global monsoon area (GMA) and (b) global monsoon intensity (GMI) for each experiment. Panels (c, d) and (e, f) show the hemispheric equivalents (HMA and HMI) for NH and SH. The ensemble mean is shown, with an 11-year running mean applied. The vertical error bars indicate the range across the five members. GMA is in Mm², where 1 Mm² = 1 × 10⁶ km².**

Both GMA and GMI show a dependence on AA forcing (Figures 4a, 4b), with a higher scaling leading to a reduction of both intensity and area. This is consistent with the effects of the scaling on global temperature, HTC and LSTC, which are also reduced at higher scalings. This dependence is clearest in GMI from 1950–1980: during this period, higher scalings produce a greater weakening of the GMI than lower scalings. This suggests a switch between GHGs and AAs dominating the influence on the monsoon from 1950–1980 across the range of uncertainty in aerosol forcing. The dependence is also clear in GMA, although the timing, duration, and strength of the GMA reduction after 1950 vary across scalings. This is most likely associated with natural variability across the five ensemble members.

Despite this variability, the effect of the scalings on GMA and GMI when averaged over 1950–2014 is also monotonic and roughly linear with scaling factor across the experiments (Table 1). The effect of the uncertainty in aerosol radiative forcing

on GMA and GMI is a reduction of 2.99% and 1.93% respectively, when increasing the scaling across its range. For context, Hsu et al. (2013) found that 1 °C of warming in CMIP5 models resulted in multi-model mean increases of 1.9% and 1.3% in GMA and GMI (see their Figure 5). The sensitivities identified here are higher (about 3.1% and 2.0% per °C), although they lie well within the range of sensitivities presented by Hsu et al. (2013).

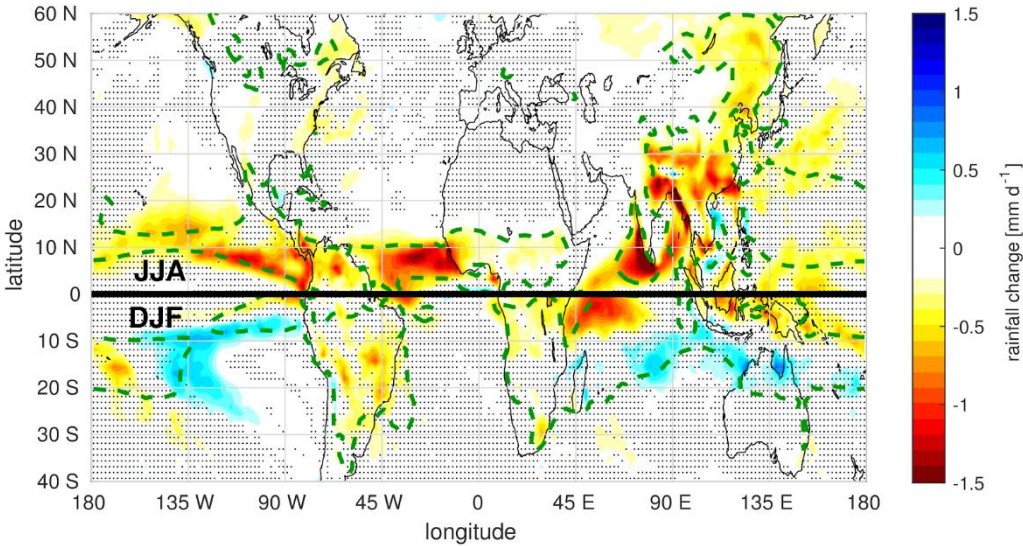

**Figure 5. The difference in monsoon rainfall (in mm d⁻¹) across the range of the scaling factors (×1.5 minus ×0.2). The summer months are shown in each hemisphere (June–August in the NH, December–February in the SH); the thick black line marks the equator. Averaged over the period 1950–2014, and across all ensemble members. The green dotted line indicates the mean GMA in the ×1.0 experiment. Spots indicate regions where the rainfall difference is insignificant with respect to variability across years and members.**

The effects of aerosol scaling on both GMA and GMI are dominated by the NH response, with a weak dependence on the scaling found in the SH (Figures 4c–4f). The effect of uncertainty in aerosol radiative forcing has substantial effects on the rainfall in the regional monsoons (Figure 5), with the greatest rainfall changes in the NH monsoons. The North American and North African monsoon experience a marked reduction, while the decrease in the Asian monsoon is even greater (consistent with the much larger aerosol emissions originating there; see Figures 1g, 1h). The effect of the scaling on the SH monsoons, in contrast, is much more variable, reflecting the much smaller local aerosol forcing. The effect of the aerosol forcing uncertainty on HMA and HMI from 1950 onwards in the NH is more than twice that in the SH (Table 1).

The sensitivity of different monsoon areas to the scaling is quantified in Figures 6 and 7. For both RMA and RMI, the greatest sensitivity to aerosol scaling lies in the East Asian and Indian sectors. Sensitivity of RMI in other regions is generally lower, while sensitivity of RMA is much smaller in all other regions except the South African sector, although this spans much of the southern Indian Ocean (Figure 1d) and is likely influenced by changing circulations and rainfall associated with the Indian monsoon during winter (Figure 5). Table 2 quantifies the differing relative contributions of changes in RMI and RMA to the overall monsoon rainfall changes across the monsoon regions. The impact of the uncertainty in scaling on the Indian and East Asian regions is a reduction (~4%) in both RMA and RMI. In the American and North African regions, the change is dominated

by the reduction in RMI; in the South African region, it is dominated by a reduction in RMA. In the Australian region, the RMI reduces but the RMA increases.

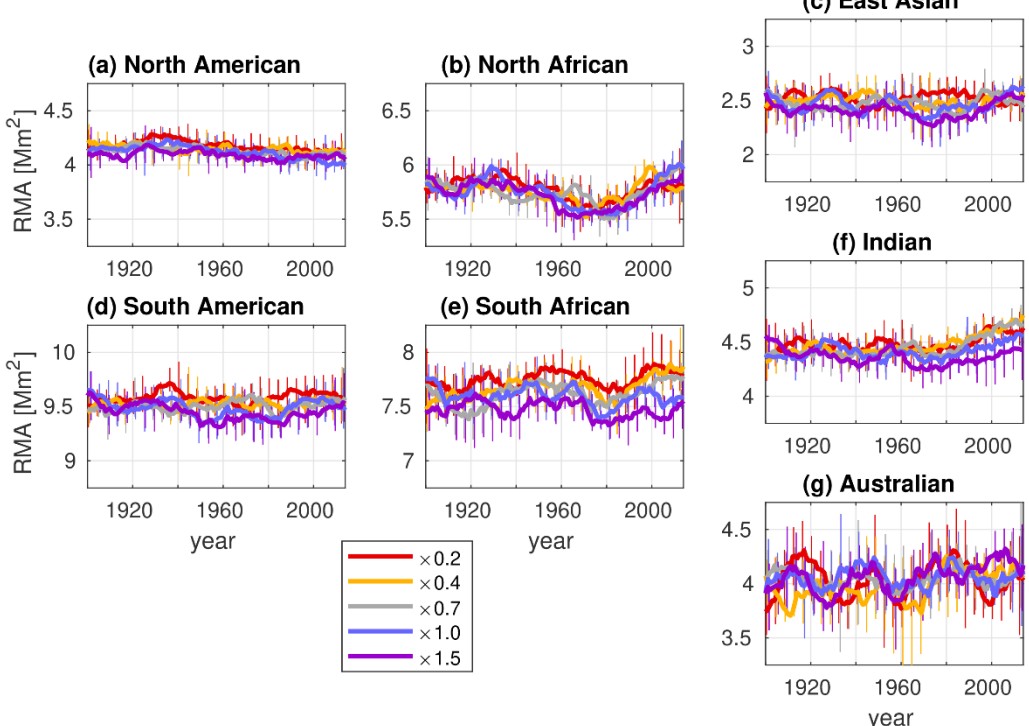

**Figure 6. Time series of the regional monsoon area (RMA), following the style of Figure 4. Here, the RMA is defined as the global monsoon area (GMA) that falls over land points within each monsoon region. Regions are as defined in Figure 1d.**

The increased sensitivity in the Asian region is consistent with studies that have shown these monsoons to be sensitive to both local and remote aerosols (Dong et al., 2016; Undorf et al., 2018). Here, large-scale monsoon changes associated with changes to the circulation are further enhanced by higher local emissions – a mechanism that is less prevalent in other monsoon regions where the aerosol burden is lower. Kitoh et al. (2013) and Lee and Wang (2014) have both examined the sensitivity of monsoon area over the Asian monsoons to changes in global temperature in CMIP5. Kitoh et al. (2013) report that the intensity of the monsoon changes to a similar extent across all monsoons, although large changes in monsoon area only occur over the Asian region and the southern Indian Ocean, which echoes our results.

While this study is the first time this particular single-model approach to understanding the effects of aerosol uncertainty on the climate system has been used, we recognise that the single-model nature of the approach could be a limiting factor. We have demonstrated that HadGEM3 performs well at representing the monsoon, and Wilcox et al (2020) have shown that HadGEM3 is one of the better models in CMIP6 at representing the rainfall and wind patterns of the Asian summer monsoon. But the question may be raised as to whether the results presented here would be applicable to other models – for example, would other model responses behave monotonically and roughly linearly with the scaling factor? To attend these questions,

we recognise the potential value of a multi-model SMURPHS-style ensemble and hence encourage modelling centres to perform similar experiments to those documented by Dittus et al. (2020).

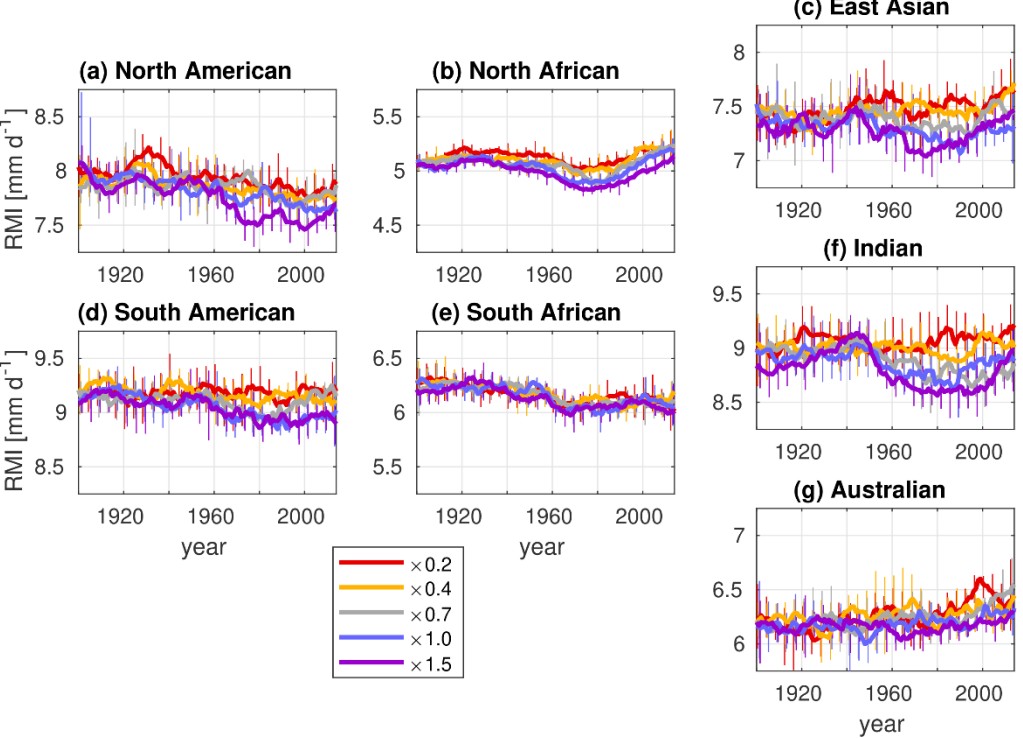

 **Figure 7. As Figure 6, but for regional monsoon intensity (RMI), defined as the total rainfall within the RMA divided by the area of the RMA.**

## 5   Summary and conclusions

The observed reduction in global monsoon area and intensity since 1950 has been widely attributed to a rapid increase in emissions of anthropogenic aerosols and their precursors. The cooling associated with these emissions is concentrated in the Northern Hemisphere, and opposes the warming effect of greenhouse gases and reduces the temperature contrast between hemispheres and between land and sea. This has been shown to weaken the monsoon circulations, resulting in a reduction of monsoon rainfall. Understanding the interplay between aerosol forcing and monsoon properties in past simulations is important in order to constrain future monsoon projections, where anthropogenic aerosol reductions are likely to strengthen the monsoon, in addition to the strengthening anticipated in response to further increases in greenhouse gases.

We explored the sensitivity of the global monsoon to uncertainty in historical aerosol radiative forcing using an ensemble of simulations in which anthropogenic aerosol and precursor emissions from 1850–2014 are scaled by five factors ranging from ×0.2 to ×1.5 (corresponding to a present-day aerosol effective radiative forcing range of −0.38 W m$^{-2}$ to −1.50 W m$^{-2}$ and

representing a large fraction of the uncertainty in present-day aerosol radiative forcing). Increasing the scaling factor from low to high results in a cooling of global temperature, a reduction of both hemispheric temperature contrast and tropical land-sea contrast, and reductions in both the global monsoon area and intensity. Across the scalings, these properties all changed monotonically and roughly linearly. When averaged over the period 1950–2014, increasing the scaling factor from ×0.2 to ×1.5 results in a 0.95 °C cooling of global temperature, a 75% reduction in hemispheric temperature contrast, a 30% reduction in land–sea temperature contrast, and a southward shift of the ITCZ by 0.28° of latitude. The global monsoon area is reduced by 3% and the intensity of the rainfall within this region is reduced by 2%. Regionally, much of the reduction in monsoon area and intensity arises in the Northern Hemisphere monsoons, particularly the Asian sector, where emission changes are greatest. Here, increasing the scaling factor from ×0.2 to ×1.5 results in reductions of monsoon area and intensity by 3.5%–5 %.

**Table 2. Values of regional monsoon area (RMA) and regional monsoon intensity (RMI), averaged over the years 1950–2014 and all five ensemble members. The difference column is the change from ×0.2 (lowest scaling) to ×1.5 (highest scaling), expressed as a percentage.**

|  | ×0.2 | ×0.4 | ×0.7 | ×1.0 | ×1.5 | Difference |
|---|---|---|---|---|---|---|
| RMA (North American) [Mm$^2$] | 4.14 | 4.13 | 4.10 | 4.08 | 4.08 | **−1.78%** |
| RMA (South American) [Mm$^2$] | 9.58 | 9.53 | 9.52 | 9.46 | 9.40 | **−1.82%** |
| RMA (North African) [Mm$^2$] | 5.73 | 5.73 | 5.73 | 5.70 | 5.66 | **−1.13%** |
| RMA (South African) [Mm$^2$] | 7.80 | 7.70 | 7.65 | 7.57 | 7.46 | **−4.16%** |
| RMA (Indian) [Mm$^2$] | 4.51 | 4.52 | 4.49 | 4.42 | 4.34 | **−3.59%** |
| RMA (East Asian) [Mm$^2$] | 2.52 | 2.48 | 2.48 | 2.46 | 2.40 | **−4.69%** |
| RMA (Australian) [Mm$^2$] | 4.04 | 4.03 | 4.03 | 4.05 | 4.10 | **+1.48%** |
| RMI (North American) [mm d$^{-1}$] | 7.87 | 7.81 | 7.82 | 7.74 | 7.63 | **−3.06%** |
| RMI (South American) [mm d$^{-1}$] | 9.19 | 9.14 | 9.08 | 8.98 | 8.98 | **−2.26%** |
| RMI (North African) [mm d$^{-1}$] | 5.12 | 5.09 | 5.05 | 5.01 | 4.94 | **−3.46%** |
| RMI (South African) [mm d$^{-1}$] | 6.11 | 6.11 | 6.10 | 6.09 | 6.06 | **−0.83%** |
| RMI (Indian) [mm d$^{-1}$] | 9.06 | 9.00 | 8.80 | 8.81 | 8.70 | **−4.00%** |
| RMI (East Asian) [mm d$^{-1}$] | 7.53 | 7.49 | 7.37 | 7.25 | 7.23 | **−4.03%** |
| RMI (Australian) [mm d$^{-1}$] | 6.33 | 6.29 | 6.29 | 6.21 | 6.16 | **−2.75%** |

Long-term monsoon variability since 1950 has very different characteristics across the scaling factors. In the ×1.5 experiment, an overall negative trend in monsoon rainfall intensity is found, dominated by strong aerosol forcing; in the ×0.2 experiment, greenhouse gases are able to dominate and monsoon intensity increases. Reducing uncertainty in the radiative forcing associated with anthropogenic aerosol would provide more reliable estimates of the future evolution of global and regional monsoons as anthropogenic aerosol and precursor emissions decline.

## Data Availability

In addition to the data used in this manuscript being available at **https://gws-access.ceda.ac.uk/public/smurphs/SMURPHS/**, this data set is part of an ongoing, larger deposit at **https://catalogue.ceda.ac.uk/uuid/5808b237bdb5485d9bc3595f39ce85e3**.

## Author Contribution

This study used data from the SMURPHS project and provided by AJD and EH. Analysis of the data was performed by JKPS, AGT, AC and LJW. The manuscript was prepared by JKPS with contributions from all co-authors.

## Competing Interests

The authors declare that they have no conflict of interest.

## Acknowledgements

This work was funded by the SMURPHS, REAL Projections, and EMERGENCE projects under the Natural Environment Research Council (NERC; Grants NE/N006054/1, NE/N018591/1 and NE/S004890/1, respectively). EH and the SMURPHS ensemble were additionally supported by the National Centre for Atmospheric Science. AC was supported by the UK Met Office Climate Science for Services Partnership (CSSP) China project funded by the Newton Fund. We acknowledge the use of the MONSooN2 system, a collaborative facility supplied under the Joint Weather and Climate Research Programme, a strategic partnership between the UK Met Office and NERC.

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
