# Peer review of "Uncertainty in Aerosol Radiative Forcing Impacts the Simulated Global Monsoon in the 20th Century"

_Atmospheric Chemistry and Physics, 2020_

## Referee Comment (RC1) · Anonymous Referee #2 · 23 Jul 2020

Review of "Uncertainty in Aerosol Radiative Forcing Impacts the Simulated Global Monsoon in the 20th Century" by Shonk et al., ACPD, 2020

This paper aims to contribute to the question of how persisting uncertainty in past emmissions of anthropogenic aerosols (AA) and their radiative forcing have impacted the climate system during the last ~100 years. To this end, they analyse the impact of AA magnitude on planetary-scale characteristics ranging from the global-mean surface temperature to monsoon area and intensity in the SMURPHS ensemble of the coupled HadGEM3-CG3.1. model. The topic is well within the scope of ACP, the paper is well-written with good graphics, and it presents a (mostly - see my ITCZ-related comment)

clear documentation of how changing the AA magnitude impacts climate. In my view, a weakness of the paper is that it very much remains at this documentary level, and gives little insight into the mechanisms that underlie the AA impact. It also limits its discussion to time-mean changes between 1950 and today, although it would seem that shorter-term changes would be more powerful to understand whether a low or high AA scaling (and thus AA radiative forcing) is more plausible. This is a long-lasting debate, and it would seem that SMURPHS could contribute here. While this somewhat limits the implications of the work, I am still in support of publication in ACP. Below are a number of smaller comments that the authors might want to consider.

1. P2, L16: Maybe cite the updated radiative forcing estimate provided by Bellouin et al. (2020, https://agupubs.onlinelibrary.wiley.com/doi/full/10.1029/2019RG000660)?

2. P2, L20: Is Fig. 1b referring to Fig. 1b of Samset et al.? I assume so, and I think this should be pointed out by writing "their Fig. 1b" (if not then in L19 China is referred to as East Asia in Fig. 1, and consistent wording would be preferrable ... but since the SSP are for the future and Fig. 1 is for the past century I assume that Fig. 1 is that of Samset et al.).

3. around P2, L25: I agree that a single model can estimate the impact of AA changes in that particular model. But a model ensemble would be needed to know if the results from the single model are plausible. This needs to be acknowledged here.

4. P2, L24: Really for the first time? I think there were other studies that played with varying aerosol emissions. E.g., the work by Dan Westervelt and colleagues (https://doi.org/10.5194/acp-18-12461-2018). This is just a study that came to my mind immediately, there are more.

5. P4, L20: I don't think standard deviation is a good measure of inter-ensemble spread for a 5-member ensemble. Why not give the range instead?

6. The model version is a development version towards the final CMIP6 version of

HadGEM3-GC3.1. Can the authors comment on how important they deem the model differences? I.e., can one combine the SMURPS ensemble with simulations with the final model version, or should the two be considered two models?

7. Fig. A1: The authors state that the figure demonstrates the fidelity of the SMUR-PHYS simulations (P4, L6). I find that hard to see from Fig. A1. The figure shows that the simulations capture the magnitude of the variability in time, but I am wondering whether they also capture some of the signals that one believes are driven by AA changes, and whether one would expect the simulations to capture such signals or not. The authors should expand on this point.

8. Fig. 2 and Tab. 1: It would be helpful if the figure and table would include the observational estimates. These are included in Fig. A1, so it should be easy to include them here as well.

9. ITCZ definition (p5, L7): I do not understand how the ITCZ is defined, and Shonk et al. (2018) did not help me either because Shonk et al. (2018) indicate a zonal-mean that does not seem to be applied here for the 50% criterion. This should be made more transparent, or, and this would seem preferable to me, the authors should consider using one of the established ITCZ defitions that are based on the precipitation centroid between 20N/S (e.g., Adam et al., 2016, doi: 10.1175/JCLI-D-15-0512.1). I would also prefer to define the ITZC based on the zonal-mean precipitation directly instead of averaging the zonal values of the ITCZ, as I would expect this to be more robust and more closely related to the hemispheric perturbations in the atmospheric energy budget introduced by the AA. This would also explain why the authors diagnose the ITCZ to be at the equator or even in the SH in the zonal-mean, which is at odds with a large body of previous work that has put the ITCZ at several degress north of the equator.

10. Definition of monsoon metrics: These include ocean areas, as can be seen in Fig. 4. I know there is some debate about whether a monsoon should be thought of to only

exist over land or not. It would help if the authors could at least briefly acknowledge this.

11. P8, L3: typo –> although they lie

---

## Referee Comment (RC2) · Anonymous Referee #1 · 31 Aug 2020

The authors present a high-level analysis of the temperature and precipitation response in monsoon regions to a wide range of aerosol emissions, scaled from the historical CMIP6 dataset. The paper is well written and the figures are clear and of high quality. The topic is squarely within the scope of the journal. However, the paper is very short, and the discussion and analysis are quite shallow. If the authors can add a little more depth to their analysis, I can recommend this for publication.

Further comments:

1. The abstract is really brief. There needs to be some detail there. The name of the model isn't even mentioned in the abstract. It appears the whole article is written more

in "Letter" format.

2. Page 1 Line 24: Not all aerosols cause negative radiative forcing

3. Page 2, Line 7-8: See also and consider citing Westervelt et al. (2018) Westervelt, D.M., A.J. Conley, A.M. Fiore, J.-F. Lamarque, D.T. Shindell, M. Previdi, N.R. Mascioli, G. Faluvegi, G. Correa, and L.W. Horowitz, 2018: Connecting regional aerosol emissions reductions to local and remote precipitation responses. Atmos. Chem. Phys., 18, 12461-12475.

4. Page 2, Line 11: Regarding AA emissions look likely to decrease. This is probably true, but nonetheless it is dependent on projections/IAMs and at the very least a citation is needed here (i.e. one of the RCP or SSP papers)

5. Page 2, Line 24: I don't believe this is the first time someone has investigated climate response to a variety of forcing levels (or emissions). Perhaps you mean the first time in this particular model.

6. Section 2. Monsoon regions (especially in Africa) may be strongly impacted by natural aerosols (dust mostly). The reader needs to know what the model is doing for dust.

7. Section 3 and Figure 2. Why not present GMST as an anomaly as in commonly done? This will make the results more comparable to the many other studies looking at temperature response to aerosols, since models may have different baseline temperatures.

8. Page 5 Line 20. "...climate responses vary monotonically and roughly linearly across the 0.2 - 1.5 scalings." I find this to be pretty interesting given the complexity and nonlinearity of the aerosol-climate system. This also may be one of the more novel findings and one that SMURPHS is uniquely positioned to answer. Perhaps this could be a sentence added to the abstract.

Page 8, line 8, final paragraph. Sorry but I don't see the point of just parachuting

in a bunch of appendix figures/tables for the other regions. Referring specifically to Figures A2, A3, and Tables A1 and A2. Any figure in the paper should be discussed and contribute to the narrative, or else it shouldn't be included. It seems that there is a wealth of interesting analysis that could be written about these two tables and two paragraphs.

---

## Author Response (AR1)

**Responses to Reviewers**

**Manuscript: "*Uncertainty in Aerosol Radiative Forcing Impacts the Simulated Global Monsoon in the 20th Century*"**

Please find our responses to the reviewers in this document. Our responses are highlighted in blue, and updated elements of the manuscript are highlighted in yellow and green in the Track Changes document included in the upload, and marked using the comment numbers in this document.

**Responses to Reviewer One**

This paper aims to contribute to the question of how persisting uncertainty in past emmissions of anthropogenic aerosols (AA) and their radiative forcing have impacted the climate system during the last ~100 years. To this end, they analyse the impact of AA magnitude on planetary-scale characteristics ranging from the global-mean surface temperature to monsoon area and intensity in the SMURPHS ensemble of the coupled HadGEM3-CG3.1. model. The topic is well within the scope of ACP, the paper is well written with good graphics, and it presents a (mostly - see my ITCZ-related comment) clear documentation of how changing the AA magnitude impacts climate. In my view, a weakness of the paper is that it very much remains at this documentary level, and gives little insight into the mechanisms that underlie the AA impact. It also limits its discussion to time-mean changes between 1950 and today, although it would seem that shorter-term changes would be more powerful to understand whether a low or high AA scaling (and thus AA radiative forcing) is more plausible. This is a long-lasting debate, and it would seem that SMURPHS could contribute here. While this somewhat limits the implications of the work, I am still in support of publication in ACP. Below are a number of smaller comments that the authors might want to consider.

> We thank the reviewer for their encouraging overall view of our submitted manuscript. While the point is well taken that AA mechanisms are not discussed in detail in this paper, on the basis that these mechanisms have been discussed extensively in previous literature we argue that their inclusion here would make the present work too long. Among previous literature (much of it cited in our introduction), there is strong evidence that the direct radiative effects and indirect cloud interaction effects of aerosol emissions have contributed at least in part to declining monsoon rainfall. At the scale of an individual region, there is evidence that both local and remotely emitted aerosols contribute to declining rainfall trends.
>
> Please see our responses to the specific questions below.
>
> Note: figure numbers in this response refer to the updated figures in the latest version of the manuscript.

**A1**. P2, L16: Maybe cite the updated radiative forcing estimate provided by Bellouin et al. (2020, https://agupubs.onlinelibrary.wiley.com/doi/full/10.1029/2019RG000660)?

> We have added this reference in the Introduction section along with their estimates. It has also been added in the full reference list at the end.

**A2**. P2, L20: Is Fig. 1b referring to Fig. 1b of Samset et al.? I assume so, and I think this should be pointed out by writing "their Fig. 1b" (if not then in L19 China is referred to as East Asia in Fig. 1, and consistent wording would be preferrable ... but since the SSP are for the future and Fig. 1 is for the past century I assume that Fig. 1 is that of Samset et al.).

> Yes, the Figure 1b referred to here is in Samset et al (2019). This has been clarified.

**A3**. around P2, L25: I agree that a single model can estimate the impact of AA changes in that particular model. But a model ensemble would be needed to know if the results from the single model are plausible. This needs to be acknowledged here.

> In a CMIP-type ensemble, uncertainty due to aerosol forcing strength alone cannot be cleanly separated from uncertainty due to structural model differences (e.g. due to different parameterisations). The strength of the SMURPHS ensemble is that we are able to cleanly quantify one of these uncertainties.  We agree that, in an ideal world, a multi-model SMURPHS-type ensemble would be a valuable addition to our datasets. The end of Section 4 already contains a recommendation that other modelling centres should perform similar SMURPHS-type simulations, and we have added more detail here highlighting that such simulations would be valuable to check that the results here are robust. We regard the findings of our manuscript as an important motivator for other modelling centres or the international community to perform a coordinated multi-model experiment. Without suitable "pathfinder" studies published in the literature, as we hope this work will become, it is difficult to launch such major multinational studies.

**A4**. P2, L24: Really for the first time? I think there were other studies that played with varying aerosol emissions. E.g., the work by Dan Westervelt and colleagues (https://doi.org/10.5194/acp-18-12461-2018). This is just a study that came to my mind immediately, there are more.

> The appearance of the phrase "first time" in this case is ambiguous as to what it points to -- indeed, there are many other studies that have scaled aerosols or set them to zero, including Westervelt et al (2018). We have rewritten this sentence to clarify what it is that was done for the first time in the SMURPHS ensemble -- that is, performing a set of historical simulations with time-varying aerosols scaled to sample a large fraction of the uncertainty in historical radiative forcing from IPCC AR5, but in a single model. We have removed the words "first time". See also a response to the other reviewer (comment B5). The reference of Westervelt et al (2018) has also been added.

**A5**. P4, L20: I don't think standard deviation is a good measure of inter-ensemble spread for a 5-member ensemble. Why not give the range instead?

> Standard deviation has been replaced with range for all of the vertical bars on all relevant figures. The corresponding captions have been updated and the text in Section 2 appended accordingly.

**A6**. The model version is a development version towards the final CMIP6 version of HadGEM3-GC3.1. Can the authors comment on how important they deem the model

differences? I.e., can one combine the SMURPS ensemble with simulations with the final model version, or should the two be considered two models?

> The two model versions differ only in their treatment of prescribed ozone concentrations. The issue is described in the Supplementary Information in Dittus et al (2020): "There is a known issue in the model version used here that causes stratospheric ozone concentrations to occur in the upper troposphere as the tropopause rises with warming, causing a small amount of unphysical warming. This issue has been resolved in the UK's contribution to CMIP6 (Andrews et al., 2020, Hardiman et al., 2019). We can compare the simulations from our ensemble with the updated versions to demonstrate that the effect of this issue is negligible over the historical period, at least for the standard scaling (Figure S1). We cannot rule out an effect for the scalings which produce larger warming levels but we expect it to be small." We have added a sentence to emphasise this.

**A7**. Fig. A1: The authors state that the figure demonstrates the fidelity of the SMURPHYS simulations (P4, L6). I find that hard to see from Fig. A1. The figure shows that the simulations capture the magnitude of the variability in time, but I am wondering whether they also capture some of the signals that one believes are driven by AA changes, and whether one would expect the simulations to capture such signals or not. The authors should expand on this point.

> Figure A1 has now been promoted to the main body of the paper, and more detail on the verification has now been added in the final paragraph of Section 2. The figure has been neatened and the bottom four panels removed for conciseness (they showed nothing that is not already clear from the GMA and GMI panels). Using this figure, we demonstrate that the model is capable of capturing aerosol-driven changes by examining the period 1950--1980, in which aerosol forcing rises. We see that the rate of change of global mean temperature and hemispheric temperature contrast vary as expected with scaling factor, with higher scaling factors leading to a more rapid decline in both properties (Figures 2a, 2b).

**A8**. Fig. 2 and Tab. 1: It would be helpful if the figure and table would include the observational estimates. These are included in Fig. A1, so it should be easy to include them here as well.

> It would be easy, although we feel it would detract from the main point of this part of the study. The emphasis in Figure 3 (also Figures 4, 6 and 7) is the impact of the scaling factor on the variables, rather than a comparison model and observations, which has been attended to already by the inclusion of Figure 2 in the main body of the paper. There are offsets between model and observations, particularly in derived rainfall quantities such as GMI and GMA -- hence observation lines on Figures 3 and 4 may lie outside the range presented. Comparison between model and observations on Figures 3 and 4 would be best represented using some sort of normalised anomaly, although this is already done in Figure 2 and would mask important features such as the sign of HTC and LSTC in Figure 3.

**A9**. ITCZ definition (p5, L7): I do not understand how the ITCZ is defined, and Shonk et al. (2018) did not help me either because Shonk et al. (2018) indicate a zonal-mean that does

not seem to be applied here for the 50% criterion. This should be made more transparent, or, and this would seem preferable to me, the authors should consider using one of the established ITCZ defitions that are based on the precipitation centroid between 20N/S (e.g., Adam et al., 2016, doi: 10.1175/JCLI-D-15-0512.1). I would also prefer to define the ITZC based on the zonal-mean precipitation directly instead of averaging the zonal values of the ITCZ, as I would expect this to be more robust and more closely related to the hemispheric perturbations in the atmospheric energy budget introduced by the AA. This would also explain why the authors diagnose the ITCZ to be at the equator or even in the SH in the zonal-mean, which is at odds with a large body of previous work that has put the ITCZ at several degress north of the equator.

> We have repeated the ITCZ location calculation using the "centroid" method of Adam et al (2016), and this has now been incorporated into Figures 2 and 3 and Table 1, replacing the Shonk et al (2018) method. The location of the ITCZ is similar using both methods, which indicates extra robustness. For reference, the Shonk et al centroid approach was applied to the global zonal mean rainfall -- this has been clarified. The location of the ITCZ in HadGEM3-GC3 varies around the equator in both definitions, which is at odds with the findings of Adam et al, but the model tends to put its ITCZ rainfall too far south (Williams et al, 2018). The text describing the ITCZ location has been updated to reflect all of this.

**A10**. Definition of monsoon metrics: These include ocean areas, as can be seen in Fig. 4. I know there is some debate about whether a monsoon should be thought of to only exist over land or not. It would help if the authors could at least briefly acknowledge this.

> We have added a statement about this in the part of the paper that introduces GMA, at the start of Section 4.

**A11**. P8, L3: typo –> although they lie

> This has been amended.

**Responses to Reviewer Two**

The authors present a high-level analysis of the temperature and precipitation response in monsoon regions to a wide range of aerosol emissions, scaled from the historical CMIP6 dataset. The paper is well written and the figures are clear and of high quality. The topic is squarely within the scope of the journal. However, the paper is very short, and the discussion and analysis are quite shallow. If the authors can add a little more depth to their analysis, I can recommend this for publication.

> We thank the reviewer for their encouraging view of our submission. We have opted to deepen some of the discussions in the main body of the manuscript, including incorporation of some of the supplementary/appendix material into the main body, and expanding detail in the results sections. Please also see our responses to specific comments below.

**B1**. The abstract is really brief. There needs to be some detail there. The name of the model isn't even mentioned in the abstract. It appears the whole article is written more in "Letter" format.

We have added more detail to the Abstract, including a mention of the name of the SMURPHS project and the model used for the simulations. We have added a little more detail on some of the key results, and mentioned the monotonic, roughly linear relationship as requested in comment B8 below. To balance this, we have also added more detail to the Conclusions.

**B2**. Page 1 Line 24: Not all aerosols cause negative radiative forcing.

The statement of negative radiative forcing here was intended to apply to the net effect across all aerosols rather than individual components -- this has now been clarified.

**B3**. Page 2, Line 7-8: See also and consider citing Westervelt et al. (2018) Westervelt, D.M., A.J. Conley, A.M. Fiore, J.-F. Lamarque, D.T. Shindell, M. Previdi, N.R. Mascioli, G. Faluvegi, G. Correa, and L.W. Horowitz, 2018: Connecting regional aerosol emissions reductions to local and remote precipitation responses. Atmos. Chem. Phys., 18, 12461-12475.

The reference has been added, although slightly below the suggested location.

**B4**. Page 2, Line 11: Regarding AA emissions look likely to decrease. This is probably true, but nonetheless it is dependent on projections/IAMs and at the very least a citation is needed here (i.e. one of the RCP or SSP papers).

Lund et al (2019) shows that aerosol emissions from three of the SSPs reduce globally over the period 2015 to 2100. We have added this reference in this sentence.

**B5**. Page 2, Line 24: I don't believe this is the first time someone has investigated climate response to a variety of forcing levels (or emissions). Perhaps you mean the first time in this particular model.

The use of "first time" here was intended to highlight the novel features of our study, but the wording did not make that clear. The novel aspect here is that we cover the historical time period since 1850 and systematically sample a large fraction of the IPCC AR5 range of aerosol forcing uncertainty (i.e., 'plausible' range of aerosol forcing). As the reviewer rightly points out, there have been many studies scaling emissions, but these have typically been idealised simulations focussed on a specific time period/region/aerosol species, so differ quite substantially in the experimental design, and have in some cases applied unrealistically large aerosol perturbations to better identify the forced response. To our knowledge, the only study to vary historical aerosol forcing through time in a similar manner is Jimenez-de-la-Cuesta and Mauritsen, 2019. However, they did not change aerosol forcing via emissions, so again it is a different experiment design. However, the wording 'for the first time' has been replaced with a new sentence to clarify the novel features. See response to comment 4 from the other reviewer.

**B6**. Section 2. Monsoon regions (especially in Africa) may be strongly impacted by natural aerosols (dust mostly). The reader needs to know what the model is doing for dust.

Mineral dust is simulated interactively in this model version using the CLASSIC aerosol module (Woodward, 2001). Changes in dust emission may arise in these simulations, associated with changes in near-surface winds and soil moisture induced by the differences in anthropogenic aerosol. This means there is the potential for a dust

feedback in these simulations, due to an induced change in the dust radiative forcing. However, the dust does not mix with the anthropogenic aerosol. This has been clarified.

**B7**. Section 3 and Figure 2. Why not present GMST as an anomaly as in commonly done? This will make the results more comparable to the many other studies looking at temperature response to aerosols, since models may have different baseline temperatures.

Global mean surface temperature has been expressed in Figure 2 and Table 1 now as anomalies with respect to the 1900--1929 mean value in the 1.0 scaling experiment.

**B8**. Page 5 Line 20. "...climate responses vary monotonically and roughly linearly across the 0.2 - 1.5 scalings." I find this to be pretty interesting given the complexity and nonlinearity of the aerosol-climate system. This also may be one of the more novel findings and one that SMURPHS is uniquely positioned to answer. Perhaps this could be a sentence added to the abstract.

This result has now been included in the Abstract, and also added to the Conclusions.

**B9**. Page 8, line 8, final paragraph. Sorry but I don't see the point of just parachuting in a bunch of appendix figures/tables for the other regions. Referring specifically to Figures A2, A3, and Tables A1 and A2. Any figure in the paper should be discussed and contribute to the narrative, or else it shouldn't be included. It seems that there is a wealth of interesting analysis that could be written about these two tables and two paragraphs.

We have shifted Figures A2 and A3 into the text (as Figures 6 and 7), and merged the tables into a single new Table 2. We have added two more paragraphs highlighting the main results and conclusions that can be drawn from these tables and figures -- primarily that the Asian monsoon regions show a stronger sensitivity of GMA to aerosol forcing than most other regions. We demonstrate that this increased sensitivity of the Asian monsoons to the scaling echoes results from CMIP5 studies, in which a warmer climate led to increases in monsoon area over Asia, yet little change in area elsewhere.

[revised manuscript text omitted]

**Commented [J3]:** Detail added

**Commented [J4]:** Recent reference added.

**Commented [J5]: B3.** Reference added.

**Commented [J6]: A1.** Reference to Bellouin et al (2020) added. The sentences in this paragraph have also been reordered to improve the flow.

**Commented [J7]: B4.** Reference added as requested to suggest that aerosol emissions look likely to decrease in the future – the study by Lund et al (2019) found aerosol emissions to reduce across the three disparate SSPs they studied.

**Commented [J8]:** Sentence added linking to recent reference.

**Commented [J9]: A2.** Clarification provided on the location of the Figure 1b referred to in this sentence – it is in the study of Samset et al (2019).

**Commented [J10]: A4/B5.** Some detail on the outline of the SMURPHS ensemble to clarify what is done here "for the first time" – and an expansion of the text about the novelty of the study.

strength of aerosol forcing, without the complications arising from structural and parametric uncertainty found in a multi-model framework. The range of model biases and aerosol process representations in a multi-model ensemble, such as the most recent phase of the Coupled Model Intercomparison Project (CMIP6), preclude the attribution of differences in the response to differences in the forcing alone. We introduce the ensemble and experimental design in more detail in Section 2. The effect of the aerosol scaling in terms of temperature contrasts across hemispheres, and between land and sea, is examined in Section 3. Section 4 presents the effects of scaling on standard metrics of the global and regional monsoons. We summarise and conclude in Section 5.

[Figure]

Figure 1. Sulphur dioxide emissions used in SMURPHS on (a) global, (b, c) hemispheric and (e–k) regional scales, in Tg yr$^{-1}$. Organic and black carbon emissions are scaled in the same way. Monsoon regions are as defined in panel (d).

**2 SMURPHS ensemble and aerosol emission data**

The SMURPHS dataset consists of historical climate simulations run over the period 1850–2014 using a fully coupled version of HadGEM3-GC3.1 at resolutions of N96 and 1° in the atmosphere and ocean respectively (Kuhlbrodt et al., 2018; Williams et al., 2018). The model version used here is a development version towards the UK submission to CMIP6 (Andrews et al, 2020), and differs only in its treatment of prescribed ozone concentrations (see Supplementary Information in Dittus et al., 2020; Hardiman et al., 2019). For the treatment of anthropogenic aerosol, HadGEM3 uses the GLOMAP two-moment aerosol scheme that includes representations of the cloud albedo and cloud lifetime effects (Mulcahy et al., 2018 and references therein). Mineral dust is simulated interactively using a bin emission scheme (Woodward, 2001). Five ensemble members are run for each of five experiments in which the historical aerosol emissions are scaled by a constant factor. This factor is applied to emissions of all species of anthropogenic aerosol and precursors, at all locations throughout the historical emissions dataset. Biomass burning emissions are included but not scaled. Five scaling factors were selected: ×0.2, ×0.4, ×0.7, ×1.0 and ×1.5, with the ×1.0 scaling corresponding to the standard CMIP6 historical protocol. The scaling factors were chosen to sample a broad range of the uncertainty in present-day aerosol radiative forcing according to Myhre et al. (2013) and Bellouin et al. (2020), and correspond to forcings of −0.38 W m$^{-2}$, −0.60 W m$^{-2}$, −0.93 W m$^{-2}$, −1.17 W m$^{-2}$ and −1.50 W m$^{-2}$ respectively. More detail on the SMURPHS ensemble is presented by Dittus et al. (2020).

The SMURPHS simulations use the same aerosol emission dataset as used in CMIP6 (Hoesly et al., 2018), which contains emissions from 1750–2014 for sulphur dioxide, black carbon and organic carbon. As an illustration of the time evolution of historical aerosol emissions, sulphur dioxide emissions from 1900 onwards are shown in Figure 1. In the early 20th century, emissions increased gradually, but then ramped up from 1950 to 1980. Since 1980, emission mitigation efforts in North America and Europe have been balanced by continued increases in Asia, causing global emissions to level off. The hemispheric asymmetry in AA emissions is clear, with the NH contributing approximately 90% of the global total throughout the 20th century (Figures 1b, 1c). Most monsoon regions show a gradual increase in emissions in the 20th century, with pronounced increases since 1970 seen in the Indian and East Asian sectors (Figures 1g, 1h). The Hoesly et al. (2018) emissions dataset is the most up-to-date inventory of historical AA emissions and is therefore considered the best estimate.

In this study, we use all five members from each of the five experiments, but include years from 1900 onwards, to allow 50 years for the model to adjust to the scalings (after Dittus et al., 2020). When considering climatological quantities, we consider the ensemble mean for each experiment to be the model estimate of the climate system under those scaling conditions and indicate uncertainty across ensemble members in terms of the range across the five members. Where quantities are averaged over areas, a cosine-based latitude weighting is applied.

**Commented [J11]: A6.** Difference between CMIP6 version of HadGEM3 and that used here explained.

**Commented [J12]: B6.** Treatment of mineral dust explained, with reference.

**Commented [J13]:** More detail added.

**Commented [J14]: A1.** Reference to Bellouin et al (2020) added here too.

**Commented [J15]:** More detail.

**Commented [J16]: A5.** Standard deviation replaced with range as requested, and updated on the affected figures.

[Figure]

**Figure 2.** Validation time series comparing the model used in the SMURPHS ensembles (HadGEM3) with observation and reanalysis datasets. For temperature quantities, we use data from 20th Century Reanalysis (20thCRA; Slivinski et al., 2019) and Hadley Centre/Climatic Research Unit Temperature (HadCRUT4; Morice et al., 2012). For rainfall quantities, we use data from the Global Precipitation Climatology Project (GPCP; Adler et al., 2003), the CPC Merged Analysis of Precipitation (CMAP; Xie and Arkin, 1997) and the Tropical Rainfall Measuring Mission (TRMM; Kummerow et al., 2000). All model ensemble members are shown, with no running means. Temperature properties span the period 1900 to 2014; rainfall properties span 1950 to 2014. Sections 3 and 4 of the paper contain descriptions of how the variables are calculated. Anomalies are calculated for each dataset and each experiment, with respect to 1961–1990 (temperature fields) and 1980–2009 (rainfall fields).

The performance of HadGEM3 at representing monsoon properties used in this study is evaluated in Figure 2. The overall trends and variability in these properties compare well between model and observations with the exception of global temperature since 2000, in which the model produces an accelerated rate of warming, although this is a recognised behaviour (Dittus et al., 2020). The model also responds reliably to changing aerosol forcing, demonstrated by the dependence of

**Commented [J17]:** A7. Figure A1 promoted to the main body of the paper; caption modified to fit. Also trimmed down to the top six panels for conciseness, as the former panels (g) to (j) did not any information not already shown in panels (e) and (f).
**A9.** ITCZ definition updated in panel (d).
[Also – Figures all renumbered to fit this figure here. Figure numbers updated in text.]

**Commented [J18]:** A7. Information about the former Figure A1 brought into its own paragraph and more detail added. We have used the changing gradients in the period 1950—1980 to demonstrate the model ability to produce the expected responses to changing aerosol forcing.

gradients in global mean temperature and hemispheric temperature contrast (HTC) in the period 1950–1980 to the magnitude of the scaling. The increasing AA emissions during this period lower global temperature, and the hemispheric asymmetry in emissions reduces the HTC. Stronger forcing scalings result in steeper declines in both properties during this period (Figures 2a, 2b). Furthermore, there is no optimal scaling factor that can reliably represent the gradients in both properties during this

5 period – lower scalings result in a more realistic decline of global mean temperature, while higher scalings (nearer ×1.0) generate a more realistic decline of HTC. Given the importance of HTC in influencing monsoon change (e.g., Bollasina et al., 2011), it is likely that the higher scaling factors will provide the most realistic representation of the global monsoon.

**3    Temperatures and contrasts**

10 The effect on global mean temperature of scaling the AA emissions is clear (Figure 3a). Higher aerosol scalings lead to cooler global temperatures, and by the 1970–2014 period there is little overlap in global temperature between scalings. We also see evidence of the control by AA emissions on the magnitude of the mid-20th-century hiatus (the period 1950–1980), in agreement with the findings of Wilcox et al. (2013) and Jones et al. (2013). The higher scalings lead to a stronger hiatus, with a global cooling of almost 0.5 °C between 1950 and 1970 in the ×1.5 experiment. The lower scalings lead to a much weaker

15 hiatus. In the ×0.2 experiment, there is only a brief departure from the positive temperature trend around 1960 and a hiatus is barely discernible. These results echo those of Dittus et al. (2020).

The hemispheric asymmetry of AA emissions leads to a much greater degree of cooling in the NH, so the strength of the forcing has a control on the hemispheric temperature contrast (HTC), defined as NH minus SH (Chang et al., 2011; Wilcox et al., 2013). Lower scalings reduce the degree of NH cooling and therefore result in a greater HTC, remaining at about 1 °C

20 from 1930 to 1990 (Figure 3b). This reflects the tendency of the NH to be, on average, warmer than the Southern Hemisphere (SH; for example, Kang et al., 2015). Under the highest scaling (×1.5), however, the HTC reduces by over 0.5 °C from 1940 to 1970, reversing in sign during the 1970s and 1980s.

This shift in HTC is reflected in the location of the ITCZ (Figure 3d). ITCZ location is determined as the latitude of the zonal mean rainfall centroid within the latitude band 20° N to 20° S, following the "centroid" method of Adam et al. (2016; "$\varphi_{cent}$"

25 in their Equation 1). Lower scalings, associated with a warmer NH and stronger HTC, lead to an ITCZ location that is further north, consistent with Hwang et al. (2013), Allen et al. (2015) and Chung and Soden (2017). The model places its ITCZ on the equator, in contrast to Adam et al.'s (2016) calculation using observational data, which places the ITCZ north of the equator. This is likely due to the tendency of the model to place its ITCZ rainfall too far south (Williams et al., 2018). Repeating the ITCZ location calculation using the method of Shonk et al. (2018) applied to zonal mean rainfall data shows a similar result.

30

**Commented [J19]:** Slight rewording to follow better from the modified end of Section 2.

**Commented [J20]:** More detail and quantification added.

**Commented [J21]:** Slight rewording, and more quantification added.

**Commented [J22]: A9.** ITCZ definition changed to reflect the figure.

**Commented [J23]: A9.** Clarification added as to why the ITCZ is near the equator rather than to the north. Comparison between Adam et al and Shonk et al methods mentioned too.

[Figure]

**Figure 3.** Time series of various atmospheric properties from the SMURPHS simulations: (a) global mean surface air temperature, expressed as an anomaly with respect to the mean value across the five members of the ×1.0 experiment in the period 1900—1929; (b) hemispheric temperature contrast (NH minus SH); (c) tropical land–sea temperature contrast, calculated in the summer months (November–March in SH, May–September in NH) for latitudes within 30° S and 30° N only; (d) global mean ITCZ location, calculated following Adam et al. (2016; see text for a description of the method). All values are ensemble means; vertical error bars indicate the range across the five ensemble members.

**Commented [J24]:** **A5.** Vertical bars have been converted from standard deviation to range. Caption modified accordingly.
**B7.** Global mean temperature lines now plotted as anomalies with respect to the 1900—1929 mean in the 1.0 scaling experiment. Caption updated accordingly.
**A9.** ITCZ definition in this plot changed to be that of Adam et al (2016). Caption updated.

Monsoon strength is also influenced by changes in the land–sea temperature contrast (LSTC), both on regional (Lau and Kim, 2017) and global (Fasullo, 2012) scales. While weaker than the effect on HTC, there is a degree of control of the aerosol scaling on the LSTC, albeit with a larger overlap between ensemble members (Figure 3c). Higher scalings result in cooler land surfaces with respect to the surrounding oceans, hence the LSTC is reduced, and the monsoon is weakened.

The control of the aerosol forcing on the properties presented in this section is demonstrated quantitatively in the top section of Table 1 in terms of means over the 1950–2014 period, when most changes in anthropogenic aerosol have occurred. All properties vary monotonically and roughly linearly across the range of scalings used in SMURPHS, with higher scalings resulting in a cooler global temperature, a weaker HTC, an ITCZ situated further south, and a weaker LSTC. The impact of the uncertainty in present-day forcing on these properties is presented in the rightmost column of Table 1 as the differences between the lowest and highest scalings (×1.5 minus ×0.2). Changing the forcing from lowest to highest value lowers global

temperature by nearly 1 °C and reduces the HTC from 1.19 °C to 0.27 °C. The zonal-mean ITCZ location shifts southwards by 0.28° of latitude, and the LSTC reduces by just over 30%, from 0.98 °C to 0.68 °C.

**Table 1. Mean monsoon-related properties, as defined in Sections 3 and 4, averaged over the period 1950–2014 (during which global aerosol emissions increased), and all five ensemble members. The difference column is the change from ×0.2 (lowest scaling) to ×1.5 (highest scaling), expressed as a percentage where indicated.**

|  | ×0.2 | ×0.4 | ×0.7 | ×1.0 | ×1.5 | Difference |
|---|---|---|---|---|---|---|
| Global mean temperature anomaly [°C] | 0.98 | 0.79 | 0.52 | 0.30 | 0.02 | **−0.95** |
| Hemispheric temperature contrast [°C] | 1.19 | 1.03 | 0.78 | 0.57 | 0.27 | **−0.91** |
| ITCZ location (latitude) [°] | 0.09 | 0.03 | −0.06 | −0.10 | −0.19 | **−0.28** |
| Tropical land–sea contrast [°C] | 0.98 | 0.92 | 0.83 | 0.77 | 0.68 | **−0.30** |
| GMA [Mm$^2$] | 126.8 | 126.0 | 125.0 | 124.3 | 122.8 | **−2.99%** |
| HMA (NH) [Mm$^2$] | 66.8 | 66.0 | 65.5 | 64.9 | 63.9 | **−4.25%** |
| HMA (SH) [Mm$^2$] | 60.0 | 59.9 | 59.5 | 59.5 | 59.0 | **−1.60%** |
| GMI [mm d$^{-1}$] | 7.76 | 7.74 | 7.69 | 7.66 | 7.61 | **−1.93%** |
| HMI (NH) [mm d$^{-1}$] | 7.68 | 7.65 | 7.57 | 7.54 | 7.48 | **−2.58%** |
| HMI (SH) [mm d$^{-1}$] | 7.86 | 7.84 | 7.82 | 7.78 | 7.76 | **−1.30%** |

**4 Monsoon area and rainfall**

We evaluate the effects of aerosol scaling on the monsoon via Global Monsoon Area (GMA) and Global Monsoon Intensity (GMI). These properties are defined following Liu et al. (2009), Hsu et al. (2011) and others: a gridbox is within the GMA if the difference in summer and winter rainfall (May to September and November to March, depending on hemisphere) is greater than 2 mm d$^{-1}$, and more than 55% of the rain falls in the summer months. The total GMA is calculated as the sum of the area of all gridboxes within the GMA region. GMI is then calculated as the total rainfall within the GMA, divided by the area of the GMA. We also define hemispheric (HMA and HMI) and regional (RMA and RMI) equivalents – these are the GMA and GMI calculated separately for either each hemisphere, or each monsoon region as defined in Figure 1d. Note that, by these definitions, monsoon regions span both land and sea. There is debate about whether monsoon indices should be defined solely over land, although here we preserve the full land-plus-sea definition of Liu et al. (2009).

**Commented [J25]: B7.** Values here converted into anomalies too.

**Commented [J26]: A9.** Numbers updated to reflect the change of ITCZ method in Figure 2.

**Commented [J27]: B9.** Regional monsoon indices introduced here in the text.

**Commented [J28]: A10.** Comment added here that the GMA metric allows for monsoons to exist both over land and sea, and that there is some debate over this.

[Figure]

**Figure 4.** Time series of (a) global monsoon area (GMA) and (b) global monsoon intensity (GMI) for each experiment. Panels (c, d) and (e, f) show the hemispheric equivalents (HMA and HMI) for NH and SH. The ensemble mean is shown, with an 11-year running mean applied. The vertical error bars indicate the range across the five members. GMA is in Mm², where 1 Mm² = 1 × 10⁶ km².

**Commented [J29]: A5.** Vertical bars previously standard deviation – they now show range. Caption modified accordingly.

5    Both GMA and GMI show a dependence on AA forcing (Figures 4a, 4b), with a higher scaling leading to a reduction of both intensity and area. This is consistent with the effects of the scaling on global temperature, HTC and LSTC, which are also reduced at higher scalings. This dependence is clearest in GMI from 1950–1980: during this period, higher scalings produce a greater weakening of the GMI than lower scalings. This suggests a switch between GHGs and AAs dominating the influence on the monsoon from 1950–1980 across the range of uncertainty in aerosol forcing. The dependence is also clear in GMA,

10   although the timing, duration, and strength of the GMA reduction after 1950 vary across scalings. This is most likely associated with natural variability across the five ensemble members.

   Despite this variability, the effect of the scalings on GMA and GMI when averaged over 1950–2014 is also monotonic and roughly linear with scaling factor across the experiments (Table 1). The effect of the uncertainty in aerosol radiative forcing

on GMA and GMI is a reduction of 2.99% and 1.93% respectively, when increasing the scaling across its range. For context, Hsu et al. (2013) found that 1 °C of warming in CMIP5 models resulted in multi-model mean increases of 1.9% and 1.3% in GMA and GMI (see their Figure 5). The sensitivities identified here are higher (about 3.1% and 2.0% per °C), although they lie well within the range of sensitivities presented by Hsu et al. (2013).

**Commented [J30]: A11.** Added "they".

[Figure]

**Figure 5. The difference in monsoon rainfall (in mm d⁻¹) across the range of the scaling factors (×1.5 minus ×0.2). The summer months are shown in each hemisphere (June–August in the NH, December–February in the SH); the thick black line marks the equator. Averaged over the period 1950–2014, and across all ensemble members. The green dotted line indicates the mean GMA in the ×1.0 experiment. Spots indicate regions where the rainfall difference is insignificant with respect to variability across years and** 10 **members.**

The effects of aerosol scaling on both GMA and GMI are dominated by the NH response, with a weak dependence on the scaling found in the SH (Figures 4c–4f). The effect of uncertainty in aerosol radiative forcing has substantial effects on the rainfall in the regional monsoons (Figure 5), with the greatest rainfall changes in the NH monsoons. The North American and North African monsoon experience a marked reduction, while the decrease in the Asian monsoon is even greater (consistent 15 with the much larger aerosol emissions originating there; see Figures 1g, 1h). The effect of the scaling on the SH monsoons, in contrast, is much more variable, reflecting the much smaller local aerosol forcing. The effect of the aerosol forcing uncertainty on HMA and HMI from 1950 onwards in the NH is more than twice that in the SH (Table 1).

**Commented [J31]:** Added "response" for clarity.

The sensitivity of different monsoon areas to the scaling is quantified in Figures 6 and 7. For both RMA and RMI, the greatest sensitivity to aerosol scaling lies in the East Asian and Indian sectors. Sensitivity of RMI in other regions is generally lower, 20 while sensitivity of RMA is much smaller in all other regions except the South African sector, although this spans much of the southern Indian Ocean (Figure 1d) and is likely influenced by changing circulations and rainfall associated with the Indian monsoon during winter (Figure 5). Table 2 quantifies the differing relative contributions of changes in RMI and RMA to the overall monsoon rainfall changes across the monsoon regions. The impact of the uncertainty in scaling on the Indian and East Asian regions is a reduction (~4%) in both RMA and RMI. In the American and North African regions, the change is dominated

**Commented [J32]: B9.** Two paragraphs added with extra detail describing the regional monsoon results from Figures 6 and 7 and Table 2.

by the reduction in RMI; in the South African region, it is dominated by a reduction in RMA. In the Australian region, the RMI reduces but the RMA increases.

[Figure]

**Figure 6. Time series of the regional monsoon area (RMA), following the style of Figure 4. Here, the RMA is defined as the global monsoon area (GMA) that falls over land points within each monsoon region. Regions are as defined in Figure 1d.**

Commented [J33]: **B9.** Figure promoted to the main part of the paper.
**A5.** Vertical bars updated to show range.

The increased sensitivity in the Asian region is consistent with studies that have shown these monsoons to be sensitive to both local and remote aerosols (Dong et al., 2016; Undorf et al., 2018). Here, large-scale monsoon changes associated with changes to the circulation are further enhanced by higher local emissions – a mechanism that is less prevalent in other monsoon regions where the aerosol burden is lower. Kitoh et al. (2013) and Lee and Wang (2014) have both examined the sensitivity of monsoon area over the Asian monsoons to changes in global temperature in CMIP5. Kitoh et al. (2013) report that the intensity of the monsoon changes to a similar extent across all monsoons, although large changes in monsoon area only occur over the Asian region and the southern Indian Ocean, which echoes our results.

While this study is the first time this particular single-model approach to understanding the effects of aerosol uncertainty on the climate system has been used, we recognise that the single-model nature of the approach could be a limiting factor. We have demonstrated that HadGEM3 performs well at representing the monsoon, and Wilcox et al (2020) have shown that HadGEM3 is one of the better models in CMIP6 at representing the rainfall and wind patterns of the Asian summer monsoon. But the question may be raised as to whether the results presented here would be applicable to other models – for example, would other model responses behave monotonically and roughly linearly with the scaling factor? To attend these questions,

we recognise the potential value of a multi-model SMURPHS-style ensemble and hence encourage modelling centres to perform similar experiments to those documented by Dittus et al. (2020).

Commented [J34]: A3. The end of the section has been expanded to cover the point that a multi-model ensemble is needed to probe the usefulness of the SMURPHS ensemble, and would be a strong forward step.

[Figure]

5 **Figure 7. As Figure 6, but for regional monsoon intensity (RMI), defined as the total rainfall within the RMA divided by the area of the RMA.**

Commented [J35]: B9. Figure promoted to the main part of the paper.
A5. Vertical bars updated to show range.

**5  Summary and conclusions**

The observed reduction in global monsoon area and intensity since 1950 has been widely attributed to a rapid increase in emissions of anthropogenic aerosols and their precursors. The cooling associated with these emissions is concentrated in the
10  Northern Hemisphere, and opposes the warming effect of greenhouse gases and reduces the temperature contrast between hemispheres and between land and sea. This has been shown to weaken the monsoon circulations, resulting in a reduction of monsoon rainfall. Understanding the interplay between aerosol forcing and monsoon properties in past simulations is important in order to constrain future monsoon projections, where anthropogenic aerosol reductions are likely to strengthen the monsoon, in addition to the strengthening anticipated in response to further increases in greenhouse gases.
15  We explored the sensitivity of the global monsoon to uncertainty in historical aerosol radiative forcing using an ensemble of simulations in which anthropogenic aerosol and precursor emissions from 1850–2014 are scaled by five factors ranging from $\times0.2$ to $\times1.5$ (corresponding to a present-day aerosol effective radiative forcing range of $-0.38$ W m$^{-2}$ to $-1.50$ W m$^{-2}$ and

Commented [J36]: Extra detail added.

representing a large fraction of the uncertainty in present-day aerosol radiative forcing). Increasing the scaling factor from low to high results in a cooling of global temperature, a reduction of both hemispheric temperature contrast and tropical land-sea contrast, and reductions in both the global monsoon area and intensity. Across the scalings, these properties all changed monotonically and roughly linearly. When averaged over the period 1950–2014, increasing the scaling factor from ×0.2 to ×1.5 results in a 0.95 °C cooling of global temperature, a 75% reduction in hemispheric temperature contrast, a 30% reduction in land–sea temperature contrast, and a southward shift of the ITCZ by 0.28° of latitude. The global monsoon area is reduced by 3% and the intensity of the rainfall within this region is reduced by 2%. Regionally, much of the reduction in monsoon area and intensity arises in the Northern Hemisphere monsoons, particularly the Asian sector, where emission changes are greatest. Here, increasing the scaling factor from ×0.2 to ×1.5 results in reductions of monsoon area and intensity by 3.5%–5 %.

**Table 2. Values of regional monsoon area (RMA) and regional monsoon intensity (RMI), averaged over the years 1950–2014 and all five ensemble members. The difference column is the change from ×0.2 (lowest scaling) to ×1.5 (highest scaling), expressed as a percentage.**

| | ×0.2 | ×0.4 | ×0.7 | ×1.0 | ×1.5 | Difference |
|---|---|---|---|---|---|---|
| RMA (North American) [Mm$^2$] | 4.14 | 4.13 | 4.10 | 4.08 | 4.08 | **−1.78%** |
| RMA (South American) [Mm$^2$] | 9.58 | 9.53 | 9.52 | 9.46 | 9.40 | **−1.82%** |
| RMA (North African) [Mm$^2$] | 5.73 | 5.73 | 5.73 | 5.70 | 5.66 | **−1.13%** |
| RMA (South African) [Mm$^2$] | 7.80 | 7.70 | 7.65 | 7.57 | 7.46 | **−4.16%** |
| RMA (Indian) [Mm$^2$] | 4.51 | 4.52 | 4.49 | 4.42 | 4.34 | **−3.59%** |
| RMA (East Asian) [Mm$^2$] | 2.52 | 2.48 | 2.48 | 2.46 | 2.40 | **−4.69%** |
| RMA (Australian) [Mm$^2$] | 4.04 | 4.03 | 4.03 | 4.05 | 4.10 | **+1.48%** |
| RMI (North American) [mm d$^{-1}$] | 7.87 | 7.81 | 7.82 | 7.74 | 7.63 | **−3.06%** |
| RMI (South American) [mm d$^{-1}$] | 9.19 | 9.14 | 9.08 | 8.98 | 8.98 | **−2.26%** |
| RMI (North African) [mm d$^{-1}$] | 5.12 | 5.09 | 5.05 | 5.01 | 4.94 | **−3.46%** |
| RMI (South African) [mm d$^{-1}$] | 6.11 | 6.11 | 6.10 | 6.09 | 6.06 | **−0.83%** |
| RMI (Indian) [mm d$^{-1}$] | 9.06 | 9.00 | 8.80 | 8.81 | 8.70 | **−4.00%** |
| RMI (East Asian) [mm d$^{-1}$] | 7.53 | 7.49 | 7.37 | 7.25 | 7.23 | **−4.03%** |
| RMI (Australian) [mm d$^{-1}$] | 6.33 | 6.29 | 6.29 | 6.21 | 6.16 | **−2.75%** |

**Commented [J37]: B1/B8.** Extra detail added in the Conclusions too, highlighting the monotonic and roughly linear changes of the monsoon variables with scaling factor.

**Commented [J38]: B9.** Table promoted to main text as Table 2. Also the two parts of this table (RMA and RMI parts) have been merged.

[revised manuscript text omitted]

**Commented [J46]:** Recent reference added.

**Commented [J47]:** B6. Reference added on mineral dust in HadGEM3.

[Figure]

Copernicus Publications
The Innovative Open Access Publisher